Polyphyly and hidden species among Hawaiʻi’s dominant mesophotic coral genera, Leptoseris and Pavona (Scleractinia: Agariciidae)

Luck Daniel G. 1 dluck@my.hpu.edu
Forsman Zac H. 2
Toonen Robert J. 2
Leicht Sarah J. 1
Kahng Samuel E. 1
1 Hawaiʻi Pacific University , Marine Science Program, Waimanalo, HI , USA
2 University of Hawaiʻi, Hawaiʻi Institute of Marine Biology , Kaneʻohe, HI , USA
Qian Pei-Yuan
Electronic publication date: 2013 Aug 13
Publication date: 2013
Volume: 1
Electronic Location ID: e132
Received 2013 Jun 3; Accepted 2013 Jul 26
Copyright: © 2013 Luck et al.
Copyright year: 2013
Copyright holder: Luck et al.
License: This is an open access article distributed under the terms of the Creative Commons Attribution License, which permits unrestricted use, distribution, and reproduction in any medium, provided the original author and source are credited.
License URL: https://creativecommons.org/licenses/by/3.0/

Keywords: Leptoseris, Pavona, Mesophotic coral ecosystems, Integrated systematics, Molecular phylogenetics, Micromorphology, cox1-1-rRNA intron, Depth zonation, Zooxanthellate corals, Taxonomy

Funding: NOAA Undersea Research Program’s HURL Grant No. NA10OAR4300132 NOAA National Marine Sanctuaries Program MOA Grant No. 2005-008/66882 Smithsonian Office of Fellowships and Internships Short-Term Visitor Award NOAA Undersea Research Program provided the funding for sample collection and scanning electron microscopy expenses. NOAA Marine Fisheries Sanctuary Program provided funding for DNA sequencing and also contributed to the salary of Z Forsman. The Smithsonian Institution’s Office of Fellowships and Internships funded a visit to the US National Museum of Natural History to examine their type collections. The funders had no role in study design, data collection and analysis, decision to publish, or preparation of the manuscript.

==============================
Widespread polyphyly in stony corals (order Scleractinia) has prompted efforts to revise their systematics through approaches that integrate molecular and micromorphological evidence. To date, these approaches have not been comprehensively applied to the dominant genera in mesophotic coral ecosystems (MCEs) because several species in these genera occur primarily at depths that are poorly explored and from which sample collections are limited. This study is the first integrated morphological and molecular systematic analysis of the genera Leptoseris and Pavona to examine material both from shallow-water reefs (<30 m) and from mid- to lower-MCEs (>60 m). Skeletal and tissue samples were collected throughout the Hawaiian Archipelago between 2–127 m. A novel mitochondrial marker (cox1-1-rRNA intron) was sequenced for 70 colonies, and the micromorphologies of 94 skeletons, plus selected type material, were analyzed. The cox1-1-rRNA intron resolved 8 clades, yet Leptoseris and Pavona were polyphyletic. Skeletal micromorphology, especially costal ornamentation, showed strong correspondence and discrete differences between mitochondrial groups. One putative new Leptoseris species was identified and the global depth range of the genus Pavona was extended to 89 m, suggesting that the diversity of mesophotic scleractinians has been underestimated. Examination of species’ depth distributions revealed a pattern of depth zonation: Species common in shallow-water were absent or rare >40 m, whereas others occurred only >60 m. These patterns emphasize the importance of integrated systematic analyses and more comprehensive sampling by depth in assessing the connectivity and diversity of MCEs.

Introduction

Within the past two decades, molecular studies have shown that traditional classification systems for stony corals (Cnidaria: Anthozoa: Scleractinia), which are based on macromorphological characters, do not accurately reflect the evolutionary relationships of many taxa (Romano & Palumbi, 1996; Romano & Cairns, 2000; Fukami et al., 2004; Fukami et al., 2008; Huang et al., 2009; Budd et al., 2010; Kitahara et al., 2010). As recently as 2008, mitochondrial and nuclear DNA revealed that a majority of scleractinian families with reef-building genera are potentially polyphyletic (Fukami et al., 2008; Huang et al., 2009). The macromorphological characters that underpin traditional scleractinian taxonomy are notoriously variable between reef habitats and biogeographic regions (Veron, 2000; Todd, 2008), leading to phenotypic overlap between distantly related taxa (i.e., homoplasy) that confounds phylogenetic analyses. However, recent studies that have integrated molecular and micromorphological/structural evidence (hereafter referred to as integrated systematic studies) have found correspondence between molecular-based and morphologically-based taxa (Fukami et al., 2004; Benzoni et al., 2010; Budd et al., 2010; Benzoni et al., 2012; Kitahara et al., 2012). Integrated systematics have resolved outstanding problems in coral taxonomy from the family-level (Budd & Stolarski, 2009) to the species-level (Forsman et al., 2010).

To date, most integrated systematic studies of hermatypic corals have focused on easily-accessible shallow-water (<30 m) species. Few species that occur primarily in mesophotic coral ecosystems (MCEs), which are coral habitats from ∼30–40 m to the deepest limits of obligate zooxanthellate corals (see Hinderstein et al., 2010), have been included in such studies. In particular, the agariciid genus Leptoseris Milne-Edwards & Haime, 1849, has not been comprehensively examined using an integrated approach despite the fact that it dominates many Indo-Pacific MCEs (Wells, 1954; Fricke & Knauer, 1986; Kahng & Kelley, 2007; Rooney et al., 2010). The most recent revision of Leptoseris by Dinesen (1980) was based largely on shallow-water material and occurred prior to the widespread use of molecular techniques in coral systematics. Since then, the genus has been included in several molecular studies (Romano & Cairns, 2000; Cuif et al., 2003; Le Goff-Vitry, Rogers & Baglow, 2004; Fukami et al., 2008; Kitahara et al., 2010), but in each case only one or two shallow-water species were used and just four of 18 described species have been studied genetically. The results of these studies have challenged the monophyly of the family Agariciidae Gray, 1847 (Fukami et al., 2008) and of the genus Pavona Lamarck, 1801 (Le Goff-Vitry, Rogers & Baglow, 2004), but none have tested the monophyly of the genus Leptoseris using more than two species. While Benzoni et al. (2012) recently provided an integrated systematic treatment of L. foliosa Dinesen, 1980, no study has applied such an approach to multiple species within the genus. Moreover, none of the species with the deepest distributions (>100 m, e.g., L. fragilis Milne-Edwards & Haime, 1849, L. hawaiiensis Vaughan, 1907 and L. scabra Vaughan, 1907) have been examined using an integrated approach. These species are particularly interesting to marine ecologists because they may have unique adaptations for efficiently capturing light (Kahng et al., 2012), allowing them to survive deeper than other photoautotrophic corals (Maragos & Jokiel, 1986; Kahng & Maragos, 2006).

While MCEs are hypothesized refugia for some reef-building species (Riegl & Piller, 2003; Bak, Nieuwland & Meesters, 2005; Bongaerts et al., 2010a), there have been few integrated systematic studies comparing shallow and mesophotic reef-building corals. The lack of studies arises from the logistical difficulties of accessing MCEs, but such studies are essential for understanding the diversity and connectivity of MCEs and assessing their conservation value. The few studies that have collected samples from MCEs have found previously unknown morphologies (Kahng & Maragos, 2006), cryptic diversity (Chan et al., 2009) and high taxonomic richness (Bridge et al., 2012). These results suggest that the global diversity of mesophotic scleractinians has been underestimated and/or the biogeographic ranges of mesophotic species are not fully explored.

The purpose of this study is to provide an integrated systematic analysis for the agariciid genera that dominate Hawaiian MCEs, namely Leptoseris and Pavona. The monophyly of these genera is tested using molecular methods, and micromorphological characters with diagnostic value are identified and evaluated in terms of their correspondence to a molecular phylogeny and traditional taxonomic designations. More reliable diagnostic characters will facilitate identification of Leptoseris and Pavona spp. in Hawaiʻi, which will enhance understanding of the ecological processes and biodiversity of Hawaiian MCEs. Accurate identification will help determine the degree of overlap between shallow-water and mesophotic species, which will facilitate investigations of the physiological adaptations that allow these corals to thrive in extreme low-light conditions.

Here, we focus on comparisons between a single novel, rapidly evolving mitochondrial marker (cox1-1-rRNA intron) and micromorphological traits. Although mitochondrial genes are often uninformative in reef-building corals (Shearer et al., 2002; Hellberg, 2006; Shearer & Coffroth, 2008), recently discovered, highly variable mitochondrial markers in Pocilloporidae (Flot & Tillier, 2007) have revealed the strongest evolutionary and biogeographical patterns (e.g., Pinzón & LaJeunesse, 2011; Keshavmurthy et al., 2013; Pinzón et al., 2013), while remaining consistent with other genetic markers. Although multiple molecular markers are often desirable in phylogenetic studies (Funk & Omland, 2003), we show here that the cox1-1-rRNA intron provides high resolution for the genera Leptoseris and Pavona with strong concordance between mitochondrial groups and discrete micromorphological characters.

Materials and Methods

Sample collections

Skeletal and tissue samples of Leptoseris and Pavona spp. were collected throughout the Hawaiian Archipelago (n = 109) and Line Islands (n = 3) (Table S1). Collections were authorized via special activity permits by the State of Hawaiʻi Board of Land and Natural Resources. Although this study focused on species from mid- to lower MCEs (i.e., >60 m), shallow-water samples (<30 m) were also collected for comparative purposes. Most samples came from MCEs in the Auʻau Channel, located between Maui and Lanaʻi, and along the nearshore shallow-water reefs off Oʻahu. Collections were made using the Hawaiʻi Undersea Research Laboratory (HURL) Pisces submersibles, technical diving or traditional SCUBA, depending on the depth. This effort included 14 submersible dives and one remotely controlled vehicle dive ranging from 49 to 216 m, 14 technical dives from 40 to 78 m, and eight shore dives from 5 to 24 m. Collected samples included all six Leptoseris spp. known from Hawaiʻi and ranged from 2 to 127 m depth, encompassing most of the accepted bathymetric range for the genus (Dinesen, 1980). (Vaughan’s (1907) record of L. hawaiiensis to 470 m is likely erroneous, see Kahng et al., 2010.) Some samples were too small for morphological analyses (n = 18); others lacked preserved tissue and were not included in molecular analyses (n = 42). The remaining 52 samples were included in both molecular and morphological analyses.

The Leptoseris type collections of the US National History Museum (USNM) and the Natural History Museum, London, UK (BNHM) were also examined (Table S2). Type specimens for a majority of Leptoseris spp. and all species previously known from Hawaiʻi were analyzed morphologically (tissues for molecular analyses were unavailable). Pavona types were not examined because at the time of museum visits Pavona was not thought to be abundant in Hawaiian MCEs.

Molecular analyses

Phylogenetic relationships were determined by analyzing sequences from the cox1-1-rRNA intron, a novel, rapidly evolving mitochondrial intergenic spacer. Preliminary analyses of cox1-1-rRNA showed that it gave much higher resolution between Leptoseris spp. than either the NAD5 5′-intron or the ITS region (DG Luck, unpublished data).

Host genomic DNA was extracted using the E.Z.N.A.® Tissue DNA Kit (Omega Bio-tek, Norcross, GA, US). The cox1-1-rRNA intron was amplified using Biomix Red (Bioline, Taunton, MA, US), a high-density PCR reaction mix, and the following primers: ZFCOXIF (5′-TCT GGT GAG CTC TTT GGG CTC T-3′) and ZFtrnar (5′-CGA ACC CGC TTC TTC GGG GC-3′). Primers were designed based on Agaricia Lamarck, 1801 and Pavona mitochondrial genomes acquired from GenBank (NC_008160,5). Each PCR totaled 12.5 µl and included reagents in the following proportions: 7.5 µl Biomix Red; 6.51 µl nanopure® (Thermo Fisher Scientific) H2O; 0.195 µl of each primer; and 0.6 µl extracted genomic DNA. The thermocycler protocol was as follows: one cycle at 95°C for 2 min; 35 cycles beginning with 95°C for 30 s, followed by 52°C for 30 s, followed by 72°C for 1 min; and one extension cycle at 72°C for 10 min. PCR products were cleaned using shrimp alkaline phosphatase and Sanger sequenced using the forward primer. Only the forward primer was necessary since it produced sufficiently long sequences (up to 900 bp) with many informative sites.

In order to provide higher resolution and to root the mtDNA phylogeny, cox1-1-rRNA sequences were obtained for Agaricia humilis Verrill, 1901, Pavona clavus (Dana, 1846) and Siderastrea radians (Pallas, 1766) from whole mitochondrial genomes available on GenBank (accession numbers: DQ643831, DQ643836, DQ643838). These three taxa were included in all phylogenetic analyses.

Sequence data were manually cleaned with BioEdit version 7.1.3 (Hall, 1999) and imported into MEGA 5.10 for alignment (Tamura et al., 2011). Separate alignments were performed using ClustalW and Muscle algorithms, first using default settings and then with reduced gap opening and extension penalties. The ClustalW alignment (with default parameters) was chosen for final analysis because it produced trees with slightly higher bootstrap support.

Maximum Likelihood (ML) and Bayesian phylogenies were constructed using evolutionary models selected by jModelTest 2.1.1 (Darriba et al., 2012). ML analysis was performed in PhylML (Guindon & Gascuel, 2003) using a transversional model with a gamma distribution of substitution rates across sites and 1000 bootstrap replicates. Bayesian analysis was performed in MrBayes 3.2.1 (Ronquist et al., 2012) using a three-parameter model with unequal base frequencies and a gamma distribution of substitution rates across sites. The Bayesian model was run over 1.75 million generations at which point the standard deviation of split frequencies fell below 0.01.

Taxonomy, sample identification, and morphological terminology

With respect to the taxonomy of the genus Leptoseris, the synonymies of Dinesen (1980) were followed except that L. tubulifera Vaughan, 1907 was treated as a valid species based on molecular and micromorphological evidence. Following the suggestion of Dai & Lin (1992), Pavona yabei Pillai & Scheer, 1976 was kept in the genus Pavona (rather than reassigned to Leptoseris as proposed by Veron & Pichon, 1980). The septal and corallite morphologies of P. yabei are more similar to those of other Pavona spp. (e.g., septocostae that always alternate in length, rows of granules rather than continuous meninae on the lateral faces of septocostae, high corallite density, predominance of radial collines, etc.) and there is no molecular evidence that it should be grouped with Leptoseris spp. All samples included in this study were identified to the species-level by comparison to type material and by consultation with taxonomists ZD Dinesen and JE Maragos.

In reference to the features of the coral skeleton, the terminology of Dinesen (1980) was employed except that ‘meninaes’ (Gill & Russo, 1980) was used rather than ‘lateral ridges’ for the paired lists or ridges along the lateral surfaces of septocostae. For detailed definitions of skeletal terms, consult Wells (1956) and Budd & Stolarski (2009).

Morphological analyses

Skeletal morphological analyses were conducted on 94 samples (52 with paired tissue vouchers) in order to (1) provide additional support for molecular-based species groups and (2) to identify diagnostic characters for species identification when molecular analyses are not possible. Emphasis was placed on identifying discrete characters because they are easily mapped onto phylogenies.

Consistent with other integrated systematic studies of scleractinians (Budd & Stolarski, 2009; Benzoni et al., 2010; Budd et al., 2010), morphological analyses focused on septocostal and costal micromorphology, although macromorphological observations of the corallum and corallites were also made. Where samples had free growth margins, costal ornamentation (‘costae’ being extensions of septocostae on the non-calicinal surface and ‘ornamentation’ being the granules/spines on costae) were also analyzed. Because of their small size (typically <100 µm in diameter) and location on the non-photosynthetic side (i.e., the underside) of the corallum, costal granules/spines were selected for closer examination since they may be subject to reduced environmental selection and therefore show less ecotypic variation within a given species.

In order to identify diagnostic characters, skeletal samples were examined under a Nikon SMZ 800 light microscope. Skeletons were bleached with sodium hypochlorite to remove tissue, rinsed and then air dried. Examinations were made first with each skeleton intact. Visible features were recorded and digital photographs of the corallites, septocostae and costae were taken using a Nikon P-6000 camera mounted to the microscope. When samples were large enough (n = 40), longitudinal sections were cut throughout the coenosteum using a Dremel® tool or wet saw to investigate the arrangement of meninaes. A subset of skeletons was also analyzed using scanning electron microscopy (SEM). Longitudinal sections or chips of these samples were mounted on metal stubs using carbon glue or tape, sputter-coated with gold-palladium for approximately 2 min, and then viewed with a Hitachi S-4800 SEM.

Results

Phylogenetic results

Sequence data from the cox1-1-rRNA intron were obtained from 70 coral colonies (Table S1) and three sequences were obtained from previously published mitochondrial genomes. The final alignment of 73 sequences was 743 bp long and included 517 variable sites, 321 of which were parsimony informative. Bayesian and ML trees had similar topologies and placed most taxa in well-supported clades (Fig. 1; only Bayesian tree shown). Pairwise nucleotide distances between significant clades were always an order of magnitude higher than within group distances (mean uncorrected p-distance of 0.193 ± 0.025 vs. 0.015 ± 0.008).

Figure 1 Bayesian phylogeny of Hawaiian Leptoseris and Pavona spp. using the mitochondrial cox1-1-rRNA intron.

Bayesian posterior probabilities (>70%) followed by maximum likelihood (ML) bootstrap support (>50%, ML tree not shown) are given at the nodes. Dashes (-) indicate statistically unsupported nodes. Roman numerals indicate clade numbers. Taxa with letters in parentheses have their micromorphologies pictured in Fig. 2.

The Bayesian phylogeny shows that the genus Leptoseris, as currently defined, is polyphyletic (Fig. 1). A majority of Hawaiian Leptoseris spp. belong to clades I–II including L. hawaiiensis, L. incrustans (Quelch, 1886), L. papyracea (Dana, 1846), L. tubulifera and L. sp. 1. However, L. scabra, which forms the sister clade (VII) to Agaricia humilis, is distantly related to clades I–II. The mean pairwise nucleotide distances between clade VII and clades I–II were among the highest distances between all groups compared (uncorrected p-distances: I vs. VII = 0.294 ± 0.018; II vs. VII = 0.284 ± 0.018). Three haplotypes of L. mycetoseroides Wells, 1954 are scattered throughout the phylogeny, but all three clearly fall outside the group formed by clades I–II.

The genus Pavona is also polyphyletic at the cox1-1-rRNA intron and consists of at least two clades (V and VIII). Clade V is sister to the group formed by A. humilis and L. scabra, whereas clade VIII is more closely related to the Leptoseris spp. of clades I–II.

Morphological results

Morphological analyses revealed that most molecular-based species groups could be independently identified using skeletal characters (Fig. 2; Table S3). Although no single character could distinguish between all Leptoseris or Pavona spp., combinations of discrete characters were effective in pairwise comparisons. Among the more important characters were the costal ornamentation, the shape and continuity of the upper septocostal margin, the development of meninaes, and species-specific macro-projections.

Figure 2 Scanning electron micrographs showing representative septocostal and costal micromorphologies for selected Leptoseris and Pavona spp.

The left-hand column gives a top-down view of the septocostae while the right-hand column shows the arrangement of the costal granulations and spines. The micrographs correspond to the taxa labeled in Fig. 1: (A) Leptoseris sp. 1; (B) L. incrustans; (C) L. hawaiiensis; (D) L. scabra; (E) Pavona? sp. 2. Scale bars are 500 µm.

Several Leptoseris spp. had clear diagnostic characters. For example, L. incrustans consistently had meninaes four to five times wider than the upper septocostal margin, which was always a narrow and continuous ridge (Fig. 2B). This width ratio was constant throughout the corallum (mean = 0.19, standard deviation = 0.09) except in the vicinity of corallites or where insertions occurred. L. tubulifera was clearly diagnosed by the presence of corallites on tubes, a characteristic absent from Pavona? sp. 2, which may also form tubes.

In cases where a species lacked a single diagnostic character, it could usually be distinguished from species with similar gross morphologies by analyzing combinations of discrete micromorphological characters. For example, L. hawaiiensis and L. scabra both grow as thin plates in lower MCEs (Figs. 3D and 3J). The costal ornamentation of the two species, however, is clearly different (Figs. 2C–2D). L. hawaiiensis has conical spines predominantly aligned in a single row down the center of each costa, whereas L. scabra has cylindrical or beady granulations in three rows, one central row and one row on each margin. Similarly, Pavona sp.1 and L. papyracea, both of which form branching colonies (Figs. 3E and 3H), are easily distinguished by their costal ornamentation. L. papyracea has exsert, conical spines arranged in a single row down the center of each costa while P. sp. 1 has triangular or bead-like granulations scattered along costae. A final example is that of L. scabra versus L. sp. 1. Both species grow as crateriform vases with irregular, bumpy surfaces in mid-mesophotic depths (∼60–80 m), but the two differ in their development of septal teeth (Figs. 2A and 2D). L. scabra has peg-like teeth formed by discontinuities of the septocostae, including their meninaes. In L. sp. 1, teeth are absent or visible only as discontinuities of the upper septocostal margin above the meninaes. L. sp. 1 also has septocostae that form numerous abortive branches at proximal cushions, with some branches turning antiparallel from their parent septocosta. In fact, L. sp. 1 is distinct from any species described in the literature and is putatively a new species in need of formal description.

Figure 3 In situ images of Leptoseris and Pavona spp. from Hawaiʻi.

Species are presented in the order they appear (from top to bottom) in Fig. 1: (A) Leptoseris sp. 1; (B) L. tubulifera; (C) L. incrustans; (D) L. hawaiiensis; (E) L. papyracea; (F) Pavona? sp. 2; (G) L. mycetoseroides (haplotype III); (H) P. sp. 1; (I) P. varians; (J) L. scabra; (K) P. maldivensis; (L) L. mycetoseroides (haplotype I). Photo credits: D Francke (B, F, K); Hawaiʻi Undersea Research Laboratory Archives, P.I. SE Kahng (A, D, H, J); R Knapstein (I); JE Maragos (C, E, G, L).

Some aspects of micromorphology were also concordant with generic differences at the cox1-1-rRNA region. For example, members of the “Leptoseris” clades (I–II) had more or less continuous meninae, whereas most Pavona spp. had granules arranged in rows along their lateral surfaces but lacked continuous meninae. The position of L. mycetoseroides outside of the clades I–II is also supported by micromorphology as all three haplotypes of L. mycetoseroides have rows of granules rather than continuous meninae. Continuous meninae, however, are apparently not synapomorphic as they are sometimes found in L. scabra (clade VII). In fact, while morphological analyses revealed many useful characters, few synapomorphies were found. Most of the patterns of costal ornamentation that differentiate between Leptoseris spp. can also be found in Pavona spp. (Fig. 2: B vs. E) and L. scabra (Fig. 2: A vs. D).

Depth distribution of Hawaiian Leptoseris and Pavona

Leptoseris and Pavona clades sorted into two major groups according to their depth distributions (Fig. 4). Most clades occurred in shallow-water (<30 m) down to mid-mesophotic depths (60–80 m). Only clades Ib (L. hawaiiensis) and VII (L. scabra) were restricted to depths >60 m. These two clades were observed deeper than any others, with colonies of L. hawaiiensis observed at 141 m and L. scabra observed to 127 m. One Leptoseris species, L. incrustans, was never observed in mesophotic depths but was among the more common agariciids observed in dimly-lit, shallow-water caves and undercuts. Of the nominal Pavona spp., only P. sp. 1 (clade V) was observed from lower MCEs (89 m); all other Pavona spp. were restricted to depths <80 m.

Figure 4 Depth distributions of major clades of the genera Leptoseris and Pavona in Hawaiʻi.

Clade coloring corresponds to that shown in Fig. 1.

Discussion

Although a previous study found polyphyly with a few samples of Pavona and Leptoseris (Le Goff-Vitry, Rogers & Baglow, 2004), this study confirms this finding with more adequate taxonomic sampling. This polyphyly can be attributed to the inclusion of L. mycetoseroides and L. scabra within the genus since both are distantly related to other Leptoseris spp. (Fig. 1) and probably in need of generic reassignment. The remaining Hawaiian Leptoseris spp. form a monophyletic group (clades I–II) that could potentially retain the name Leptoseris, depending on the phylogenetic position of the type species, L. fragilis.

This study is also the first to use the cox1-l-rRNA intron as a phylogenetic marker for scleractinians. In Leptoseris and Pavona, this intron is highly variable and capable of distinguishing between closely related species. Such variability is atypical of anthozoan mtDNA, which normally evolves too slowly to distinguish between congenerics (Shearer et al., 2002; Hellberg, 2006). Since the cox1-l-rRNA intron gave high resolution across several agariciid genera, it may be useful for resolving species-level relationships throughout the family Agariciidae.

Because mtDNA-based trees may not accurately reflect phylogenetic descent (Funk & Omland, 2003; McGuire et al., 2007), many phylogenetic studies use nuclear-DNA (nDNA) to corroborate mtDNA-based trees (Romano & Cairns, 2000; van Oppen, Koolmes & Veron, 2001; Fukami et al., 2008). However, analysis of several nuclear markers is often necessary to achieve the same level of confidence that gene and species trees are congruent that could be achieved with a single mitochondrial marker (Moore, 1995). Morphological characters, provided they are rigorously defined (e.g., discrete), can provide an alternative data set for corroborating phylogenies inferred from mtDNA (Puorto et al., 2001; Wiens & Penkrot, 2002; Rubinoff & Holland, 2005). The strong correspondence found here between mitochondrial cox1-1-rRNA groups and discrete skeletal characters suggests that the mtDNA tree accurately reflects the species trees for these genera.

In Leptoseris and Pavona, micromorphology (sensu Budd & Stolarski, 2009) generally showed higher correspondence with molecular data than macromorphology, which was highly convergent. The colony form of L. scabra, for instance, resembles that of both L. hawaiiensis and L. sp. 1 at their respective depths yet L. scabra is not closely related to either species (Fig. 1). In both cases, L. scabra is easily distinguished using micromorphological characters. The correspondence between micromorphological and molecular evidence is consistent with the results of recent works in scleractinian systematics. Benzoni et al. (2012) found that Craterastrea levis, which had been synonymized with L. foliosa based on affinities in corallum morphology, differed from L. foliosa in micromorphology, microstructure and at several molecular markers. Kitahara et al. (2012) found that the microstructure of Dactylotrochus cervicornis (Moseley, 1881) agreed with cox1 data that had, surprisingly, placed it within the Agariciidae.

Of the micromorphological characters explored in this study, costal ornamentation is of particular diagnostic importance for Leptoseris and Pavona spp. Differences in septal ornamentation (e.g., granule shape) have been found to be effective at distinguishing between Atlantic and Pacific faviids and mussids (Budd & Stolarski, 2009). The diagnostic utility of costal ornamentation for unifacial agariciids is consistent with those results because costal and septal ornamentation are homologous; like interspecific differences in septal ornamentation, differences in costal ornamentation reflect differences in the development and arrangement of calcification zones (i.e., rapid accretion front vs. thickening deposits, see Stolarski, 2003). For corals growing in the lower mesophotic where useable light is unidirectional (Kirk, 1994), costal ornamentation (on the shaded side of a colony) theoretically does not affect colony photophysiology and should show less ecotypic variation than septal ornamentation (on the side of the colony exposed to light), potentially enhancing the diagnostic value of costal relative to septal ornamentation.

In contrast to micromorphological characters, many macromorphological (i.e., gross, colony-level appearance) characters in Leptoseris and Pavona are not concordant with molecular results. The three haplotypes of L. mycetoseroides, for example, all have the collines (protuberant ridges between corallites) that ostensibly define the species (Wells, 1954; Dinesen, 1980), yet these haplotypes are scattered throughout the mtDNA phylogeny. Another example is that of Pavona sp. 1 versus L. papyracea, two species that clearly differ in their costal ornamentation (see results section). P. sp. 1 is common in museum collections but is almost always misidentified as L. papyracea because it also has a branching growth form. Mitochondrial data, however, agree with the observed differences in costal ornamentation and confirm that the two species are distinct (Fig. 1).

The ambiguous boundaries between Leptoseris and Pavona in traditional taxonomic systems may have contributed to the narrative that Leptoseris spp. dominate lower MCEs in the Indo-Pacific (Vaughan, 1907; Wells, 1954; Rooney et al., 2010). The results of this study show that in Hawaiʻi Leptoseris spp. coexist with Pavona spp. and at least one other clade (which contains L. scabra) at depths >80 m. The collection of P. sp. 1 from 89 m surpasses the previous global depth record for the genus Pavona (73 m, Gardiner, 1905), and the collection of P. varians from 73 m in the Auʻau Channel extends the Hawaiian depth record for the species from a previous mark of 48–53 m (Vaughan, 1907). These new depth records for Pavona spp., along with the discovery of one putative new Leptoseris sp. and several undescribed morphologies (Table S3), indicate that scleractinian diversity in lower Hawaiian MCEs has been underestimated (e.g., Kahng & Kelley, 2007; Rooney et al., 2010). As the most remote archipelago in the world, Hawaiʻi is characterized as a low diversity region and Hawaiian coral fauna are depauperate relative to that of the Indo-West Pacific (IWP) (Grigg, 1983; Grigg, 1988). The discovery of putative new species and expanded vertical distributions of Hawaiian mesophotic corals suggests that the diversity of IWP MCEs, which are relatively unexplored (Kahng et al., 2010) but occur in a region of high shallow-water diversity, may be substantially greater than previously estimated.

The zonation of Leptoseris and Pavona spp. from shallow (2 m) to deep-water (≥153 m, see Kahng & Maragos, 2006) in Hawaiʻi offers the opportunity to study the processes of niche diversification, speciation and possibly adaptive radiation within a group of sympatric marine species. Adaptive radiations can occur when a group of organisms acquires a key innovation that allows them to use an underutilized set of resources (Schluter, 2000). For some Leptoseris and Pavona spp., highly efficient light absorption (see Kahng et al., 2012) may have enabled colonization of deeper habitats (i.e., different niches) where lower levels of useable light previously prevented other species from surviving or competing for space. Divergent selection between habitats could have eventually led to reproductive isolation and then speciation (Schluter, 2000). Divergence in habitat depth seems to have driven speciation in Hawaiian limpets (Cellana spp.) (Bird et al., 2011) and in eastern Pacific rockfishes (Ingram, 2011). In scleractinian corals, a growing number of studies have pointed to ecological divergence as an explanation for partitioning of species or genotypes by depth (Carlon & Budd, 2002; Vermeij & Bak, 2003; Bongaerts et al., 2010b). Divergent selection between shallow and mesophotic habitats may have contributed to the diversification of Hawaiian Leptoseris and Pavona and could explain their zonation along depth gradients.

Molecular data at two mitochondrial markers now suggest that taxonomic revisions of the genera Leptoseris and Pavona are warranted (Le Goff-Vitry, Rogers & Baglow, 2004; this study). The strong correspondence between mtDNA and micromorphology found here suggests that these revisions should focus on micromorphological characters. The development of more reliable diagnostic characters for Hawaiian agariciids will enable comparative studies along depth gradients. Accurate species identification will facilitate comparisons between mesophotic communities in Hawaiʻi versus other regions and help determine if the pattern of depth zonation observed in Hawaiʻi exists in higher diversity systems such as the IWP.

Supplemental Information

Table S1 Inventory of samples analyzed.

Samples with values in the column ‘cox1-1-rRNA clade’ were included in molecular phylogenetic analyses. HPU = Hawaiʻi Pacific University; NR = not recorded.

Click here for additional data file.

Table S2 Type material examined from the US National Museum of Natural History (USNM) and the Natural History Museum, London, UK (BM).

The synonymies proposed by Dinesen (1980) are followed except that L. tubulifera T.W. Vaughan 1907 is treated as a valid species.

Click here for additional data file.

Table S3 Matrix of selected macro- and micromorphological characters for Hawaiian Leptoseris and Pavona spp.

Diagnostic or highly informative characters are in bold. Character codes are explained in the footnotes below the table.

Click here for additional data file.

Hawaiʻi Undersea Research Laboratory provided the submersibles and crew used to collect a majority of samples; D Wagner and JE Maragos provided additional material. SD Cairns supervised a visit to USNM to view the type collections stored there. BMNH loaned type material for examination. ZD Dinesen and JE Maragos gave taxonomic consultation on the genus Leptoseris. D Wagner provided feedback on the manuscript. K McClanahan and MM Mocaer helped collect morphometric data and analyze skeletons.

Additional Information and Declarations

Competing Interests

Author Contributions

Field Study Permissions

DNA Deposition

The authors declare no competing interests.

Daniel G. Luck conceived and designed the experiments, performed the experiments, analyzed the data, wrote the paper.

Zac H. Forsman conceived and designed the experiments, edited and reviewed manuscript.

Robert J. Toonen contributed reagents/materials/analysis tools, edited and reviewed manuscript.

Sarah J. Leicht performed the experiments, analyzed the data.

Samuel E. Kahng conceived and designed the experiments, contributed reagents/materials/analysis tools, edited and reviewed manuscript.

The following information was supplied relating to ethical approvals (i.e., approving body and any reference numbers):

Board of Land and Natural Resources, State of Hawaiʻi issued SAP 2011-30 and SAP 2011-34. A majority of the corals, however, were actually collected in US Federal Waters and did not require a permit to collect.

The following information was supplied regarding the deposition of DNA sequences:

GenBank accession numbers:

Accession #’s species; KF437720 Leptoseris hawaiiensis; KF437721 Leptoseris hawaiiensis; KF437722 Leptoseris hawaiiensis; KF437723 Leptoseris scabra; KF437724 Leptoseris scabra; KF437725 Leptoseris scabra; KF437726 Leptoseris scabra; KF437727 Leptoseris sp. 1; KF437728 Leptoseris sp. 1; KF437729 Leptoseris scabra; KF437730 Leptoseris scabra; KF437731 Leptoseris scabra; KF437732 Leptoseris scabra; KF437733 Leptoseris sp. 1; KF437734 Leptoseris sp. 1; KF437735 Leptoseris sp. 1; KF437736 Leptoseris hawaiiensis; KF437737 Leptoseris hawaiiensis; KF437738 Leptoseris hawaiiensis; KF437739 Leptoseris sp.; KF437740 Leptoseris scabra; KF437741 Leptoseris sp. 1; KF437742 Pavona sp. 1; KF437743 Pavona sp. 1; KF437744 Pavona sp. 1; KF437745 Pavona sp. 1; KF437746 Leptoseris sp. 1; KF437747 Pavona varians; KF437748 Leptoseris scabra; KF437749 Leptoseris hawaiiensis; KF437750 Leptoseris hawaiiensis; KF437751 Leptoseris hawaiiensis; KF437752 Leptoseris hawaiiensis; KF437753 Leptoseris scabra; KF437754 Pavona varians; KF437755 Pavona varians; KF437756 Leptoseris sp. 1; KF437757 Leptoseris incrustans; KF437758 Leptoseris incrustans; KF437759 Leptoseris incrustans; KF437760 Pavona? sp. 2; KF437761 Pavona cf. explanulata; KF437762 Pavona sp.; KF437763 Leptoseris tubulifera; KF437764 Pavona? sp. 2; KF437765 Pavona? sp. 2; KF437766 Leptoseris incrustans; KF437767 Leptoseris incrustans; KF437768 Leptoseris mycetoseroides; KF437769 Leptoseris tubulifera; KF437770 Leptoseris scabra; KF437771 Leptoseris sp. 1; KF437772 Leptoseris tubulifera; KF437773 Pavona varians; KF437774 Pavona sp. 1; KF437775 Leptoseris mycetoseroides; KF437776 Leptoseris mycetoseroides; KF437777 Leptoseris mycetoseroides; KF437778 Leptoseris papyracea; KF437779 Pavona duerdeni; KF437780 Pavona maldivensis; KF437781 Leptoseris papyracea; KF437782 Leptoseris hawaiiensis; KF437783 Leptoseris hawaiiensis; KF437784 Leptoseris hawaiiensis; KF437785 Pavona? sp. 2; KF437786 Pavona varians; KF437787 Leptoseris hawaiiensis; KF437788 Leptoseris mycetoseroides; KF437789 Pavona varians.

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
