# Peer review of "Polyphyly and hidden species among Hawaiʻi’s dominant mesophotic coral genera, Leptoseris and Pavona (Scleractinia: Agariciidae)"

_PeerJ, doi:10.7717/peerj.132_

## Round 0.1 · original submission · Minor Revisions

Although reviewers found the merit of this manuscript, they raised a number of issues that have to be carefully addressed. For instance, the writing has to be substantially improved and the student's supervisor shall certainly edit the ms carefully and thoroughly before resubmission. The reference citation has to be adequate and updated as well as precise. A better introduction (and shorter) to the topic is need; Results section shall state findings only rather than having materials that shall be a part of M & M or Discussion. Discussion shall highlight the key findings and then discuss meaning of those findings.... The authors are advised to submitted a revision ms together with a detailed response to the comments made by each reviewer (preferred to be in a format of point-to-point).

·

Basic reporting

The manuscript is well written and provides important taxonomic information on species of the genus Leptoseris and Pavona, thus I strongly believe the article should be accepted for publication. Please find some minor comments below.

Title: "Polyphyly, cryptic species and depth zonation...", I think "hidden species" is a better term then "cryptic species", considering your phylogeny recovers very well the identified morphological groups.

Abstract: "...One new Leptoseris species...", please change to "...One putative new Leptoseris species..." as it is not formally described here. However, I strongly encourage the authors to do so in an appropriate journal, e.g. Zootaxa.

Line 24: “…Leptoseris H. Milne-Edwards & J. Haime, 1849…”, it seems to me rather unusual to use the initials of the authors of the first descriptions, please remove. Also, considering your work has a strong taxonomical focus, you may consider to list the references of the first descriptions in the reference list.
Line 35, 38, 40, 41, etc: See comment above.

Line 35: “…Agariciidae Gray, 1947…”, this should be “…Agariciidae Gray, 1847…”

Section “Morphological analyses” (Line 139). Please stress again that you investigated 52 samples here.
Line 157: “When samples were large enough…” please name how many samples you included here.

Line 177: “…clade (VII) to A. humilis, is distantly…” The genus name should be written out when used for the first time in a new section, i.e. methods, results, etc.

Figure 1. and Figure 4.: Change the colour of either species II or species VII as these are too similar and hard to differentiate.

Experimental design

The study is well executed.

Validity of the findings

The findings are well presented.

·

Basic reporting

This is a very interesting study that has been conducted in a rigorous, quantitative manner. The one area that requires tightening before publication is the language, which is not always clear and in certain places could be more concise. There are many short paragraphs which should be combined with other paragraphs throughout the manuscript. I will provide more detailed comments below on these matters, but apart from writing style this manuscript is ready for publication and provides important information on the phylogenetics of the Scleractinia.

Experimental design

Research questions are clearly defined and investigated using multiple lines of evidence. The methods are well described and could easily be replicated by the reader if necessary. Although I am not familiar with the specific genetic technique used, the authors clearly explain their reasons for their choice, which seems to be broadly applicable to other equivalent studies.

Validity of the findings

The genetic evidence is generally consistent with micromorphology, which 1) suggests the authors conclusions are robust; and 2) fits with similar studies of other Scleractinian taxa. The results are well explained, and the discussion is adequate and relevant.

Additional comments

Most of my specific comments are of a minor nature.
Abstract:
I would remove the section about refugia from the abstract. You don't really discuss refugia in the Discussion, so I would also leave it out here. The refugia issue could be addressed in the discussion. While there is evidence of depth partitoning, most of these species still have a pretty wide depth range; species occurring to 40 m are still likely to be able to utilise deep refugia from most disturbances (provided there is connectivity among deep and shallow populations).

Introduction:
Line 14 - The second paragraph should be combined with the first.

With regards to writing style, the language could be more concise and flow more effectively in some places. You often start your sentences with phrases that don't add anything to the sentence, and this should be avoided. Rather than saying things like "It is often thought that corals occur...", just say "Corals occur..." This is a common theme through the manuscript which needs to be addressed. The subject matter is very interesting and you have done a great job to work all this stuff out, so it would be a shame if the manuscript was not written clearly and concisely.

For example, the second paragraph (line 14) starts with "These findings have ushered in a new era of coral systematics that de-emphasizes macromorphology"...

This could be written much more clearly. I would suggest something like:
Traditionally, coral taxonomy has been conducted using macromorphology (e.g. Veron AIMS monographs), which is notoriously variable among reef habitats and biogeographic regions. However, recent studies have demonstrated good congruence between molecular and micromorphological characters, resolving numerous problems in coral systematics".


Similarly, the third paragraph starts with "As with much coral reef science, however, these efforts...". This sentence could be written more clearly and concisely, such as " To date, most research has focused on species occurring on easily-accessible shallow-water reefs less than 40 m deep."

Line 27: Never start a sentence with brackets, and the paragraph probably needs to be restructured to incorporate your definition of MCEs.

Line 132: L. yabei is generally regarded as Leptoseris by most people. This may not be true, but I think you need to say why you went with Pillai and Scheer rather than Veron, as most people will be more familiar with Veron's work.

Line 152: "more free to vary" - clarify what you mean by this.

Line 160 - combine this paragraph with the previous.

Line 170. This is where you need to cite Fig. 1. "Both the Bayesian and ML trees showed similar topologies, with most taxa falling into well-supported clades (Fig. 1; only Bayesian tree shown). Pairwise.....than within group distances (information). You don't need the last sentence (The Baysenian Tree....). You can use the Fig. caption to say why you only showed the Bayesian Tree and not the ML.

Line 175: What do you mean by the "main" Leptoseris clades? Do you mean the clades currently regarded as being in the genus Leptoseris? Or clades you have defined?

Line 196: "L. tubulifera was clearly identified by the presence of corallites on tubes, a characteristic absent from P. sp2, which may also form tubes".

Line 200: "In cases where a species...."

Line 232: 'were collected', not 'was collected'

Line 232: 'bathymetric distribution' rather than 'vertical distribution'

Line 243 - incorporate this sentence into the pr3evious one - "This polyphyly can be attributed to the inclusion of L. myc and L. scabra within the genus, since both are only distantly related to other Leptoseris and probably in need of generic reassignment".

Line 258: You clearly show that the current taxonomy based on macromorph. is no good, but are there other macro characteristics that may be more helpful, even if we have to look more closely. After all, we need to have some way of assessing things in the field!

Line 267-273. These sentences are not clearly written. I understand what you're saying but they need to be clarified.

Line 312: No need to start sentences with phrases like "in fact". "Divergence among depths may have driven speciation...."

Reviewer 3 ·

Basic reporting

The “Materials and Methods” and “Discussion” sections are currently too wordy (i.e. repetitive and excessive detail provided) and should be rewritten in a more concise fashion. This would greatly enhance the readability of the manuscript.

Experimental design

The aim of the study is not reported in a consistent manner: mostly it is presented as a “revision of the Hawaiian Leptoseris” although this varies across title, abstract and introduction (e.g. whether it includes Pavona). Given the focus of the manuscript on MCEs and inclusion of data on both Leptoseris and Pavona, would it be an idea to consistently present the aim as “to provide an integrated systematic analysis for the Agariciid species that dominate Hawaiian MCEs.”? Maybe the inclusion of Pavona could be reflected in the title as well?

Validity of the findings

Although I think that in the case of this manuscript it is acceptable that only a single mitochondrial marker was used, the authors should clearly acknowledge and discuss the various limitations of a single-marker/mitochondrial approach.

Additional comments

The manuscript by Luck et al. presents an interesting dataset and a timely genetic/micromorphometric evaluation of the deep-water Leptoseris and Pavona species from around Hawaii. I would like to see this manuscript published, however there are several minor concerns that I would like to see addressed first.

The one thing that is really lacking is a key (or table with identifying characters only) that offers the reader a morphological guideline to identifying the different Leptoseris and Pavona species based on macro- and, where necessary, micro-morphological characters.

SPECIFIC MINOR COMMENTS:
TITLE: I think that given the lack of quantitative information on depth zonation in this study that the “depth zonation” aspect should not be as prominently featured in the title.
ABSTRACT: “its inaccessible habitat” – please rephrase as it implies that (1) MCEs are not accessible at all and (2) Leptoseris only occurs at mesophotic depths.
ABSTRACT: “…to compare obligate zooxanthellate corals from shallow-water habitats to lower MCEs (2-127 m)” – too broad, rephrase statement or specify that this study is focused around Leptoseris and Pavona only.
ABSTRACT: “Leptoseris and other Agariciids”, given that only Leptoseris and Pavona were collected it may better to just state that.
INTRODUCTION: Please combine the first 2 paragraphs.
INTRODUCTION (L23): “The mesophotic analogue to many dominant shallow-water genera, the agariciid genus Leptoseris …” – please rephrase
INTRODUCTION (L27) “MCEs are coral habitats from ~30-40m …” – this statement should not be in parentheses, and should probably be featured at the start of this paragraph.
INTRODUCTION (L55): The study would probably be better introduced with a statement like “The purpose of the study is to provide an integrated systematic analysis for the Agariciid species that dominate Hawaiian MCEs.”
INTRODUCTION (L58): maybe expand with “evaluated in terms of their correspondence to a molecular phylogeny and traditional taxonomic species designations.”
RESULTS (L187): “most species… could be independently identified using skeletal characters” – clarify if this is “species” according to traditional taxonomic designations or based on genetic clade.
RESULTS (section c): I do not really think this belongs in the Results, but instead in section (a) of the Materials and Methods. It details sample collection details and contextual distribution information rather than quantitative information about depth distributions.
DISCUSSION (L296): Maybe include reference to earlier papers mentioning the low scleractinian diversity of Hawaiian MCEs. Also – better to replace “suggest” with “indicate”.
DISCUSSION (L310-318): In addition to the cited studies that have looked at these evolutionary processes in non-related taxa, it may be good to cite the several studies that have addressed habitat-driven divergence in scleractinian corals, e.g. the work on the genus Madracis and the Montastrea annularis species complex in the Caribbean, and Seriatopora in the Indo-Pacific.

---

## Round 0.2 · accepted · Accept

Authors have been able to address most of the concerns raised by the reviewers and the revised submission is much improved. I am happy to accept it if all the figure quality and writing style can pass through technical checks of technical team.